

# Dynamics in benthic community composition and influencing factors in an upwelling-exposed coral reef on the Pacific coast of Costa Rica

Ines Stuhldreier[1,2], Celeste Sánchez-Noguera[1,3], Florian Roth[1,2], Carlos Jiménez[3,4], Tim Rixen[1], Jorge Cortés[3] and Christian Wild[1,2]

[1] Leibniz Center for Tropical Marine Ecology, Bremen, Germany
[2] Faculty of Biology and Chemistry, University of Bremen, Bremen, Germany
[3] Centro de Investigación en Ciencias del Mar y Limnología, Universidad de Costa Rica, San Pedro, San José, Costa Rica
[4] Energy, Environment and Water Research Center, Cyprus Institute, Nicosia, Cyprus

Corresponding author
Ines Stuhldreier,
ines.stuhldreier@uni-bremen.de

## ABSTRACT

Seasonal upwelling at the northern Pacific coast of Costa Rica offers the opportunity to investigate the effects of pronounced changes in key water parameters on fine-scale dynamics of local coral reef communities. This study monitored benthic community composition at Matapalo reef (10.539°N, 85.766°W) by weekly observations of permanent benthic quadrats from April 2013 to April 2014. Monitoring was accompanied by surveys of herbivore abundance and biomass and measurements of water temperature and inorganic nutrient concentrations. Findings revealed that the reef-building corals *Pocillopora* spp. exhibited an exceptional rapid increase from 22 to 51% relative benthic cover. By contrast, turf algae cover decreased from 63 to 24%, resulting in a corresponding increase in crustose coralline algae cover. The macroalga *Caulerpa sertularioides* covered up to 15% of the reef in April 2013, disappeared after synchronized gamete release in May, and subsequently exhibited slow regrowth. Parallel monitoring of influencing factors suggest that *C. sertularioides* cover was mainly regulated by their reproductive cycle, while that of turf algae was likely controlled by high abundances of herbivores. Upwelling events in February and March 2014 decreased mean daily seawater temperatures by up to 7 °C and increased nutrient concentrations up to 5- (phosphate) and 16-fold (nitrate) compared to mean values during the rest of the year. Changes in benthic community composition did not appear to correspond to the strong environmental changes, but rather shifted from turf algae to hard coral dominance over the entire year of observation. The exceptional high dynamic over the annual observation period encourages further research on the adaptation potential of coral reefs to environmental variability.

# INTRODUCTION

Coral reef benthic communities are controlled by abiotic and biotic drivers, such as waves, water depth, reef habitat, temperature, nutrients and herbivory (*Littler & Littler, 1984*; *Kleypas, McManus & Meñez, 1999*; *Gove et al., 2015*). Anthropogenic disturbances (e.g., herbivore removal or nutrient and sediment input) may affect these drivers and decouple biophysical relationships between environmental and biotic conditions (*Williams et al., 2015*). Long-term background physical conditions drive spatial patterns in benthic community composition (*Done, 1992*; *Hughes et al., 2012*). Although it is also recognized that also low- and high-frequency variations in environmental conditions can alter coral reef benthic community composition and productivity (*Leichter, Stewart & Miller, 2003*; *Gove et al., 2015*), it is not well known which parameters are most influential in driving benthic community composition on a fine temporal scale.

To understand the effects of variable environmental drivers on coral reef ecosystems it is useful to monitor ecosystems with a pronounced natural temporal variability in conditions, such as upwelling regions. Transport of subthermocline water to the surface (in the form of coastal upwelling, large amplitude internal waves (LAIW) and internal bores) acts to decrease sea water temperature and increase nutrient concentrations in reef waters on different temporal scales ranging from minutes to several days (*Leichter & Miller, 1999*; *Leichter, Deane & Stokes, 2005*; *D'Croz & O'Dea, 2007*; *Schmidt et al., 2012*). Previous studies in areas exposed to upwelling or LAIW mainly focused on spatial differences in benthic communities comparing exposed versus sheltered sites (*Glynn & Stewart, 1973*; *Roder et al., 2011*; *Schmidt et al., 2012*). Studies on temporal differences in benthic community composition in response to upwelling are scarce (but see *Eidens et al., 2015*). In general, observations on a high temporal scale are often lacking in coral reef ecology. By conducting weekly observations over an entire year of monitoring, the present study aimed to advance the understanding of fine-scale dynamics in coral reef community composition, particularly in response to seasonal upwelling conditions.

We selected the Gulf of Papagayo at the northern Pacific coast of Costa Rica to investigate the question of how annual variability in environmental parameters affects coral reef communities. A topographic depression in the lowlands of southern Nicaragua and northern Costa Rica allows strong winds to blow across from the Caribbean during the northern hemisphere winter (*McCreary, Lee & Enfield, 1989*; *Amador et al., 2006*; *Willett, Leben & Lavín, 2006*). On the Pacific side, this leads to the displacement of superficial waters away from the coast, which causes the shallow thermocline to break the surface (*Fiedler & Talley, 2006*). This seasonal upwelling can decrease mean seawater temperatures from around 28 °C between May–November to 23 °C between December–April (*Jiménez, 2001a*; *Alfaro et al., 2012*). During pronounced upwelling events, water temperatures can drop by 8–9 °C (*Alfaro & Cortés, 2012*). Low temperatures are accompanied by decreases in pH and oxygen concentration (*Rixen, Jiménez & Cortés, 2012*) as well as peaks in inorganic nutrient concentrations, which seasonally promote the growth of fleshy macroalgae (*Fernández-García et al., 2012*; *Cortés, Samper-Villarreal & Bernecker, 2014*).

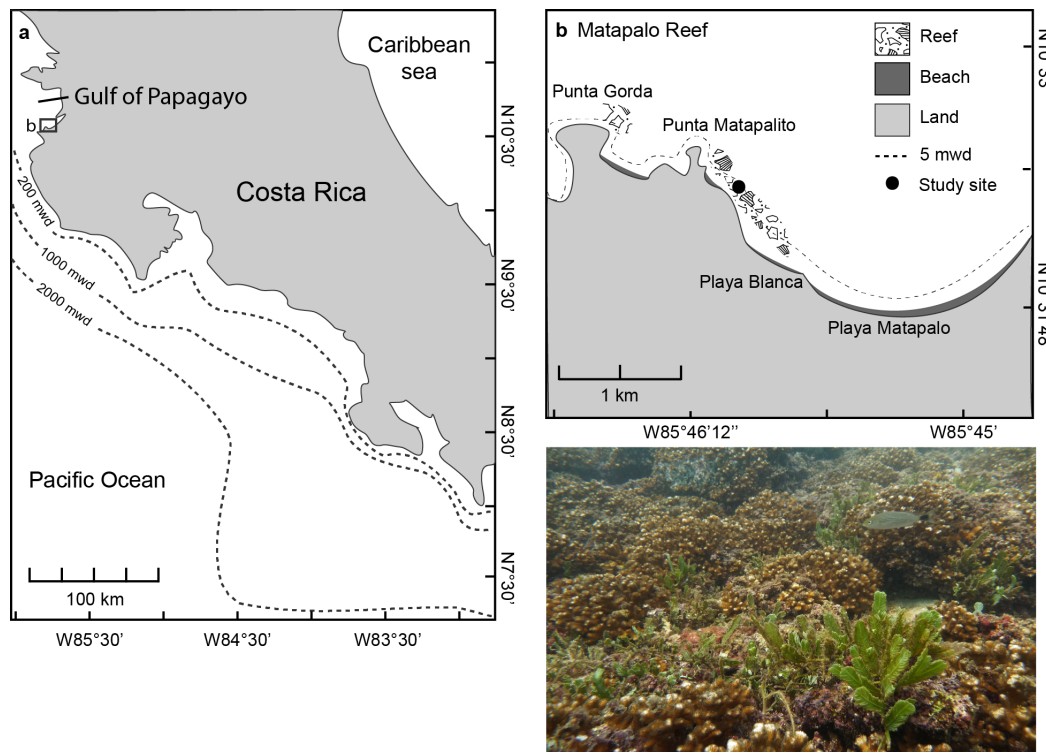

**Figure 1** **Location of the Gulf of Papagayo (A) and Matapalo reef (B).** The photograph was taken at the study site indicated in (B) at 5 m water depth (mwd) and shows the reef structure and dominant benthic organisms.

The present study monitored the temporal variation in a reef community exposed to moderate upwelling in a weekly resolution over one year. Observations of benthic community composition were accompanied by monitoring of herbivores (sea urchin abundance and fish biomass) and key water parameters (temperature and inorganic nutrient concentrations). The objectives of the study were to (1) observe the dynamics in benthic community composition over one year, (2) identify the influencing biological and (3) environmental factors and (4) develop an ecological perspective for local coral reefs. Important related key questions were: (i) which benthic groups benefit from seasonal changes in biotic and abiotic parameters? (ii) which parameter has the strongest effect on benthic community composition? and (iii) does the benthic community follow a seasonal cycle?

# METHODS

## Study site

The study was conducted at Matapalo reef, located 12 km south of Culebra Bay in the Gulf of Papagayo at the northern Pacific coast of Costa Rica from April 2013 to April 2014 (Fig. 1A). The reef framework at Matapalo in 3–9 m water depth is built by branching corals of the genus *Pocillopora* and stretches 1.2 km from Punta Matapalito to the west end of Playa Matapalo (Fig. 1B). Field work was conducted under permits issued by the National System of Conservation Areas (SINAC) of Costa Rica (permit No: 019-2013-SINAC).

## Monitoring of benthic community composition

Monitoring took place on a reef patch of around 600 m$^2$ (10.539°N, 85.766°W), 500 m north–west of Playa Blanca (study site in Fig. 1B). Quadrats of 50 × 50 cm ($n = 5$) were permanently marked with iron stakes and repeatedly observed for changes in benthic coverage every week. The location of permanent quadrats was chosen by placing a 50 × 50 cm frame alternately left and right every 2 m along a transect line parallel to the coast in 5 m water depth. For observation, the PVC frame with a 5 × 5 cm grid was placed over the stakes as a reference, and live coral, dead coral, crustose coralline algal (CCA), turf algal, macroalgal, sand, rubble and sessile benthic invertebrate cover on the substrate was quantified from directly above the grid using SCUBA. The weekly quantification was done on a relatively small area of the reef, but results were supported by monthly chain surveys along transects covering the whole reef patch. Transects of 10 m length ($n = 5$) were permanently marked with iron stakes parallel to the coast in 5 m water depth in about 3 m distance to the permanent quadrats. A 10 m iron chain with 532 links was laid out over the reef substrate along each transect, and seafloor coverage was observed under each link. Proportional seafloor cover was calculated by relating the average number of links in each category to the total number of links, and rugosity was calculated by dividing the horizontal distance covered by the chain ($d$) by its real length ($l$) (rugosity index: RI = $1 - d/l$).

## Monitoring of herbivores

Surveys of sea urchin and fish abundances were conducted monthly between 9:00 and 14:00 on all five permanent transects using SCUBA. Sea urchins of the species *Diadema mexicanum* and *Eucidaris thourasii* were counted in 1 m belts on both sides of the transect lines (total survey area = 100 m$^2$) from April 2013 to April 2014. Fish surveys were conducted from November 2013 to March 2014. During surveys, all individuals (excluding cryptic species) within 2.5 m belts on both sides of the transect lines (total survey area = 250 m$^2$) were identified to species level and assigned to size classes (5–10 cm, 10–15 cm, 15–20 cm, 20–25 cm, 25–30 cm, 30–35 cm and 35–40 cm). Multiple swims ($n = 3$) with intervals of 5 min between swims were conducted on each transect in order to increase the precision of data for each transect. Fish species were categorized into trophic groups according to their dominant food source (herbivores, planktivores, invertebrate feeders and predators) according to descriptions on FishBase (*Froese & Pauly, 2012*). Biomass of species and trophic guilds was calculated from abundance, mean length of each size class and species- or family-specific Bayesian length–weight relationship parameters available on FishBase. Shannon's diversity index ($H$) and Evenness ($E_H$) was calculated for each survey day from the number and relative contribution of each fish species to total fish abundance counted on all 5 transects.

## Monitoring of water parameters

Monitoring of benthic community composition was accompanied by measurements of temperature with a Manta 2 Water Quality Multiprobe deployed directly on the reef

substrate (recording over 1–6 h in 4 min intervals) and weekly determination of inorganic nutrient concentrations. Water samples for nutrient measurements were taken in triplicate from directly above the reef surface within glass jars (500 mL), filtered immediately through disposable syringe filters (pore size 0.45 μm) and stored cool for transportation. Ammonia ($NH_4^+$) was determined fluorimetrically within 24 h after sampling with a Trilogy® Laboratory Fluorometer/Photometer (Turner Designs) according to *Holmes et al. (1999)* and *Taylor et al. (2007)* (detection limit (LOD) = 0.023 μmol L$^{-1}$). Determination of phosphate ($PO_4^{3-}$) was conducted spectrophotometrically with the same device following the standard protocol of *Murphy & Riley (1962)* (LOD = 0.033 μmol L$^{-1}$). Sub-samples were kept dark and frozen until the end of the study period and were analyzed for nitrate ($NO_3^-$) and nitrite ($NO_2^-$) concentrations using a Thermo Scientific UV Evolution 201® photometer based on a method revised by *García-Robledo, Corzo & Papaspyrou (2014)* ($LOD_{(NO_2^-)}$ = 0.151 μmol L$^{-1}$; $LOD_{(NOx)}$ = 0.162 μmol L$^{-1}$). A detailed description of the temporal variability in physicochemical and organic parameters at the study site can be found in *Stuhldreier et al. (2015)*.

## Data analyses

Differences in fish and sea urchin abundances over time were tested with one way ANOVAs followed by Tukey post-hoc tests in SigmaPlot13 for Windows. Correlations between cover types (*Pocillopora* spp., CCA, *C. sertularioides*, turf, sand, rubble, cyanobacteria) and between sea urchin abundances and turf algal cover were tested via Pearson Product Moment Correlation in SigmaPlot 13. Temporal dynamics in environmental parameters and benthic community composition were examined with Principal Coordinate Ordinations (PCO) (*Gower, 1966*) and tested by Permutation Multivariate Analyses of Variance (PERMANOVA) (*Anderson, 2001*; *Anderson, Gorley & Clarke, 2008*) in PRIMER 6. Prior to analysis, weekly means of environmental parameters (temperature, $PO_4^{3-}$, $NH_4^+$, $NO_3^-$) were normalized and a resemblance matrix calculated using Euclidean similarity. The resemblance matrix of benthic community (cover of *Pocillopora* spp., dead coral, CCA, turf algae, *C. sertularioides*, sand, rubble and zoanthids) was calculated using Bray Curtis similarity. The fixed factors used for the analysis were (1) upwelling period with two levels (upwelling: February and March 2014, and non-upwelling: remaining months) and (2) individual months with 13 levels (each month from April 2013 to April 2014). To test how well patterns in community composition correlated to patterns in environmental parameters, the function RELATE in PRIMER 6 was used, which tests matching of the resemblance matrices by Spearman Rank correlation. Values in the text are reported as mean ± standard error (SE) if not stated otherwise.

## RESULTS

### Benthic community shifts from turf algae to hard coral dominance

At Matapalo reef, areas of dead reef structure overgrown with turf algae alternate with healthy reef areas dominated by live coral cover. At the time of study, the benthic community in healthy reef areas was dominated by the sole reef building corals *Pocillopora*

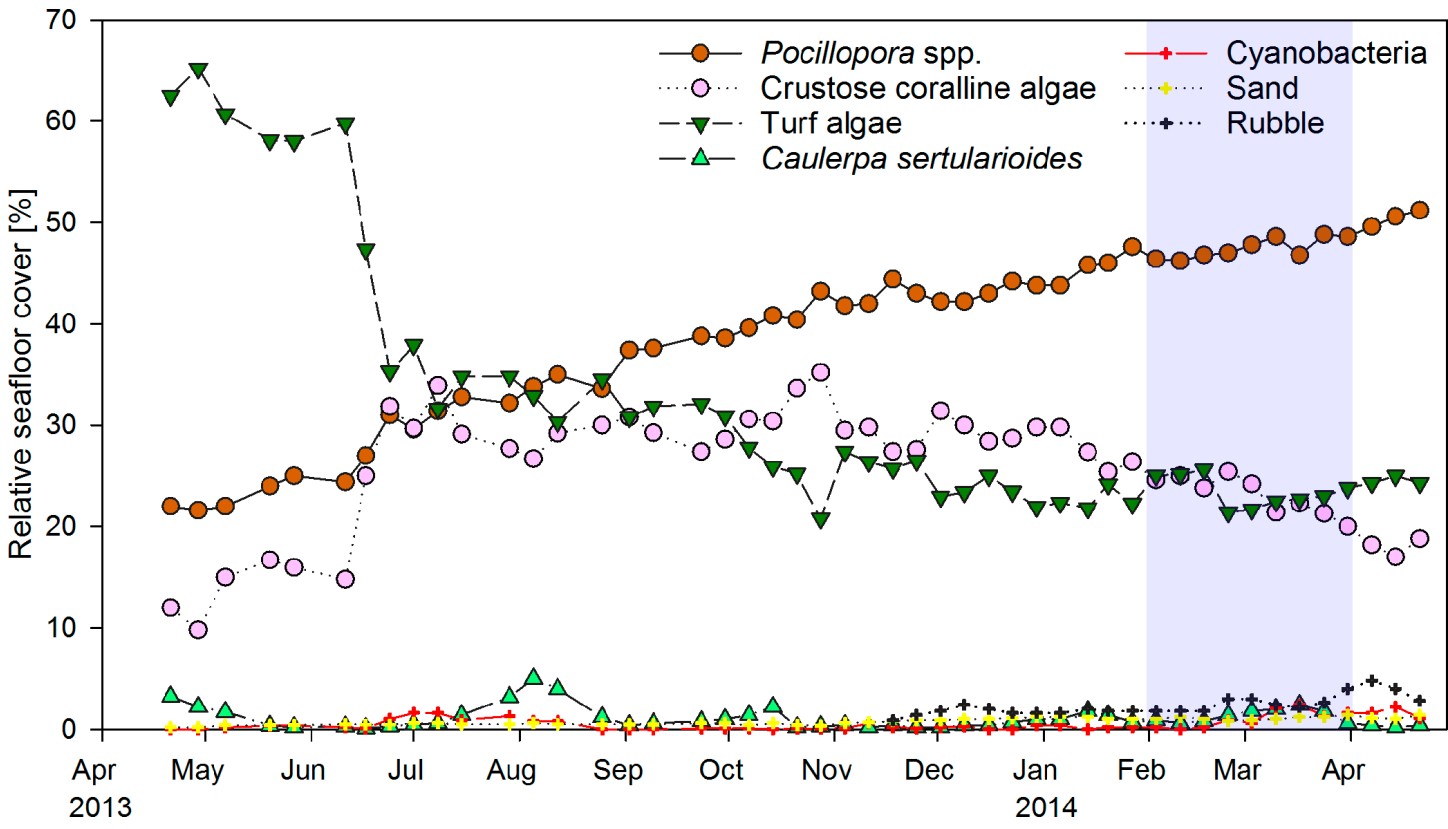

**Figure 2  Temporal changes in coral reef community composition.** Displayed is the mean proportional cover of benthic organisms or substrates in permanent quadrats of 50 × 50 cm ($n = 5$) from April 2013 to April 2014 in a weekly resolution. Shading indicates the period of strongest upwelling in February and March (see Fig. 4).

spp. (*P. elegans* and *P. damicornis*), crustose coralline algae (CCA) and turf algae. Turf algae communities existed as fine tufts of filamentous algae and cyanobacteria or in denser conglomerates with *Dictyota* spp. The green algae *Caulerpa sertularioides* was the only individually growing fleshy macroalga on the reef and covered large parts of the dead reef framework in shallow areas and on the edges of reef patches. Single coral colonies of *Pocillopora* spp., *Pavona* spp. and *Psammocora* spp. and coral rubble could be found on sandy patches between the reef rock areas. The rugosity of the reef substrate was low with values between 0.08 and 0.16.

Weekly monitoring of permanent quadrats revealed a major shift from turf algae to hard coral dominance within the year of observation (Fig. 2). Cover of the hard coral *Pocillopora* spp. increased continuously from 22% in April 2013 to 51% in April 2014. Turf algae initially covered around 60% of the substrate, but decreased to 35% within two weeks in June 2013. The drop in turf algae cover resulted in an increase in the relative cover of CCA from 15 to 30%, as previously turf algae had overgrown much of the CCA-covered substrate. CCA cover remained around 30% until January 2014, when it decreased slightly coincident with an increase in coral cover. The macroalga *Caulerpa sertularioides* initially covered 3% of the substrate in the permanent quadrats in April 2013, but decreased to

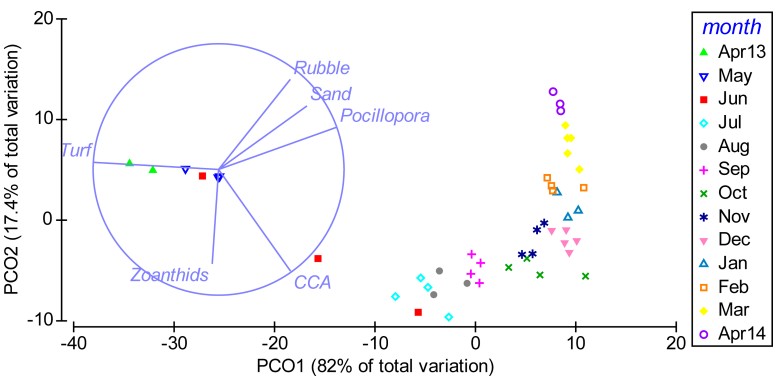

**Figure 3 Temporal shift in benthic community composition.** Weekly sampling data from April 2013 to April 2014 were grouped by the factor month. The distance between data points reflects their similarity in benthic community composition (close = similar) and the shift along axes can be assigned to changes in benthic cover types (arrows). PCO1 correlates positively to cover of *Pocillopora* spp. (Pocillopora) (Pearson correlation, $r = 0.94$), sand ($r = 0.70$), rubble ($r = 0.57$) and crustose coralline algae (CCA) ($r = 0.57$) and negatively to turf algae (Turf) ($r = -0.99$). PCO2 correlates positively to rubble ($r = 0.71$) and sand ($r = 0.51$) and negatively to CCA ($r = -0.81$) and zoanthids ($r = -0.75$). Only variables with $r > 0.5$ are displayed.

0.1–0.5% by May 2013. Macroalgae subsequently exhibited temporary increases in cover of up to 5% in August 2013 and 2.5% in March 2014. Percent cover of *Pocillopora* spp. and turf algae in permanent quadrats correlated significantly to their respective coverages in the monthly chain transects covering the whole reef patch (Pearson correlation, $n = 13$, $r = 0.848$ and $0.692$, $P < 0.001$ and $P = 0.009$, respectively). By contrast, correlations of CCA and *C. sertularioides* coverages between the quadrats and chain transects were not significant ($n = 13$, $r = 0.540$ and $0.461$, $P = 0.057$ and $0.113$, respectively), due to the fact that the cover of *C. sertularioides* was higher towards the peripheries of the reef patch. The drop in *C. sertularioides* cover from April to May 2013 could therefore be observed in a more pronounced way in the chain transects across the whole reef patch (decrease from 15 to 1% cover, Fig. 4A). The decrease in *C. sertularioides* cover was associated with whitening of the macroalgal's stolons and fronds. Sand and rubble in permanent quadrats increased from around 0.5% substrate cover between April–October 2013 to 1.4% and 3–4% respectively in March and April 2014. Zoanthids were the only sessile invertebrates observed (up to 0.2% cover). Cover of turf algae in permanent quadrats correlated negatively with that of *Pocillopora* spp. (Pearson correlation, $r = -0.914$), CCA ($r = -0.613$) sand ($r = -0.674$) and rubble ($r = -0.537$, all $P < 0.001$, $n = 48$). Sand and rubble also correlated positively with each other ($r = 0.834$) and with the cover of *Pocillopora* spp. ($r = 0.818$ and $r = 0.756$ respectively, both $P < 0.001$, $n = 48$) and cyanobacteria ($r = 0.311$, $P = 0.033$ and $r = 0.444$, $P = 0.002$ respectively, both $n = 48$).

Multivariate analyses further illustrated the pronounced shift in benthic community composition over the study period (Fig. 3). From April to October 2013 the community shifted along PCO1, which explained 82% of the total variance in the data and was positively correlated with cover of *Pocillopora* spp., and negatively with turf algae. From October 2013 to April 2014, the composition shifted further along PCO2, which explained

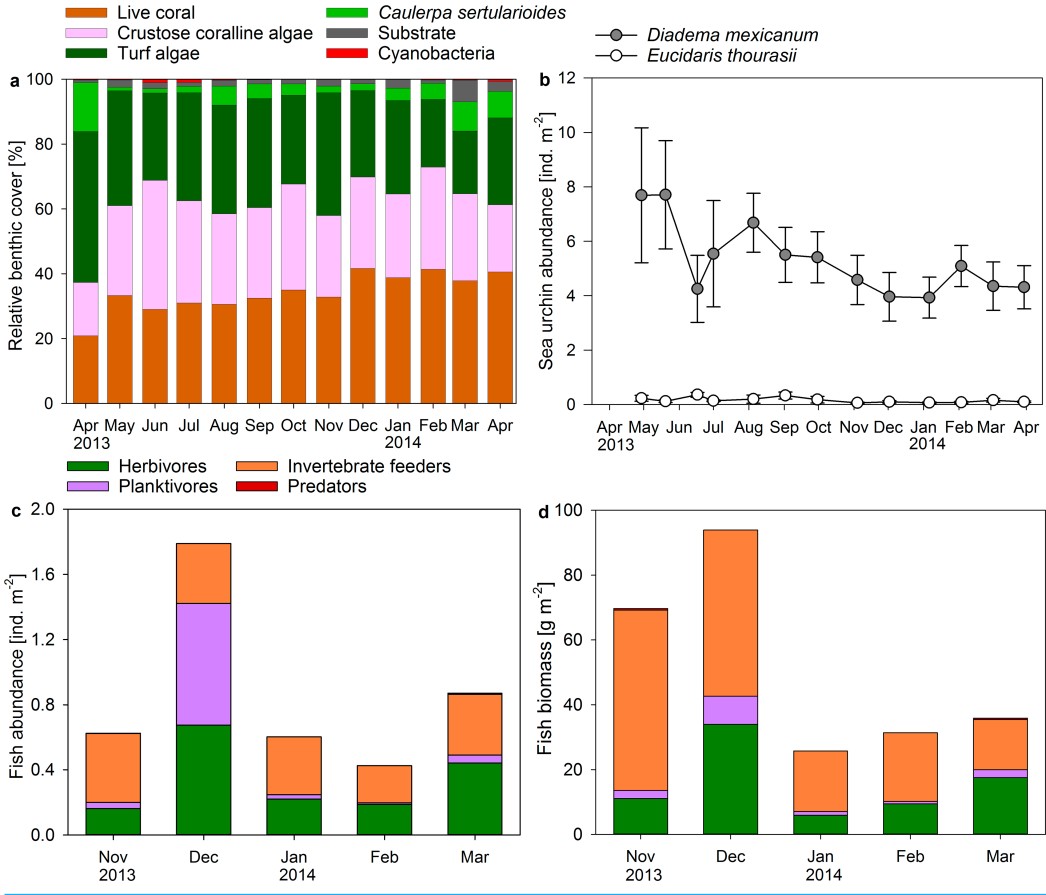

**Figure 4 Changes in coral reef benthic, sea urchin and fish communities.** Displayed are (A) mean proportional cover of benthic organism along permanent chain transects ($n = 5$), mean abundances of (B) sea urchins and (C) fish, and (D) fish biomass calculated from abundances and mid length of size classes in permanent belt transects ($n = 5$) from April 2013 to April 2014 in a monthly resolution.

17.4% of the total variance and showed a positive correlation to rubble and a negative correlation to CCA. Differences in community composition were significant between months (PERMANOVA, Pseudo-$F(12, 34) = 47.848$, $P$(perm) $= 0.001$, perms $= 998$).

## Reef exhibits high abundances of herbivorous sea urchins and parrotfish

Sea urchin abundances in the reef averaged $5.31 \pm 0.36$ ind. m$^{-2}$ for *Diadema mexicanum* and $0.16 \pm 0.03$ ind. m$^{-2}$ for the pencil sea urchin *Eucidaris thourasii* (Fig. 4B). Individuals of *Astropyga pulvinata* and *Tripneustes depressus* occasionally occurred on the reef but were rarely counted during surveys. Abundances of *D. mexicanum* and *E. thourasii* did not change significantly during the study period (ANOVA, $F(12, 52) = 0.991$, $P = 0.470$ and $F(12, 52) = 1.336$, $P = 0.228$, respectively). Abundances of *D. mexicanum* correlated positively with turf algae cover in permanent quadrats (Pearson correlation, $r = 0.787$, $P = 0.001$, $n = 13$) and on benthic transects ($r = 0.676$, $P = 0.011$, $n = 13$).

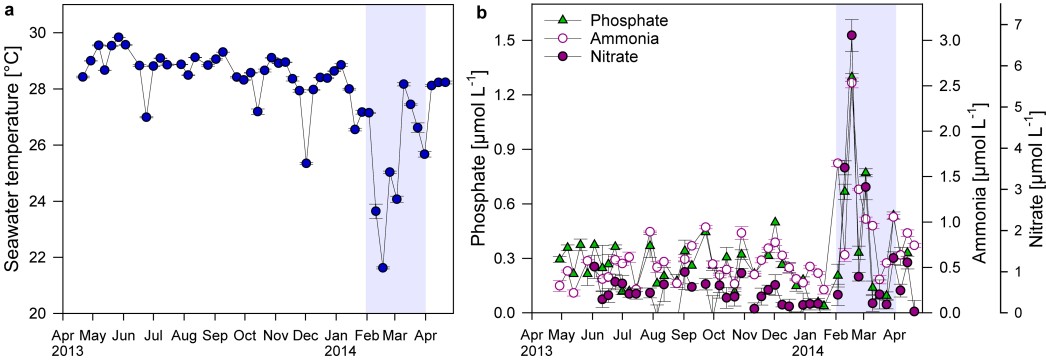

**Figure 5 Changes in temperature and nutrient concentrations.** Displayed are mean ± SE values for (A) water temperature (measured over 1–6 h in 4 min intervals) and (B) nutrient concentrations ($n = 3$) directly above the reef substrate from April 2013 to April 2014 in a weekly resolution. Shading indicates period of strongest upwelling in February and March.

The Shannon–Wiener diversity of the observed fish community ranged from $H = 1.7$ in March 2014 to $H = 2.3$ in November 2013 and was positively related to the evenness of species distribution ($E_H = 0.59$–$0.78$). Small groupers were the only predators recorded during fish surveys at Matapalo reef. Invertebrate feeders (mainly pufferfish, triggerfish and butterflyfish) dominated total abundance and biomass of fish except in December 2013, when high abundances of planktivorous *Chromis atrilobata* dominated fish counts, and in March 2014, when many large individuals of the herbivorous parrotfish *Scarus ghobban* were present (Figs. 4C and 4D). Herbivores were comprised of territorial damselfish and roving herbivorous parrotfish, while surgeonfish were sometimes observed on the reef during diving, but hardly counted during surveys. Total fish abundance on the reef was significantly elevated in November 2013 (ANOVA, $F(4, 20) = 6.095$, $P = 0.002$) and total biomass was significantly elevated in November and December 2013 (ANOVA, $F(4, 20) = 14.729$, $P < 0.001$). A statistical correlation of fish abundance or biomass with benthic community composition was not conducted due to the low sample size of fish data ($n = 5$).

## Upwelling causes high variability in water temperature and inorganic nutrient concentrations

Short drops in temperature of approximately 2 °C occurred at Matapalo reef in June, October and December 2013. More pronounced and longer lasting cooling events, coupled with increased nutrient concentrations, occurred between February and March 2014 (Figs. 5A and 5B). Average water temperatures during this upwelling period decreased from $28.5 \pm 0.1$ °C to $26.2 \pm 0.6$ °C, while average $PO_4^{3-}$, $NH_4^+$ and $NO_3^-$ concentrations increased by 70, 80 and 270%, respectively. During strongest upwelling events, mean daily seawater temperatures dropped by 7 °C to minima of 22 °C and nutrient concentrations increased 5-, 4- and 16-fold to maxima of $1.3\ \mu mol\ PO_4^{3-}\ L^{-1}$, $2.5\ \mu mol\ NH_4^+\ L^{-1}$ and $6.7\ \mu mol\ NO_3^-\ L^{-1}$, respectively.

According to the PCO analysis, the variability in environmental parameters at Matapalo from April 2013 to April 2014 was mainly explained along PCO1 (79.3% of total variation),

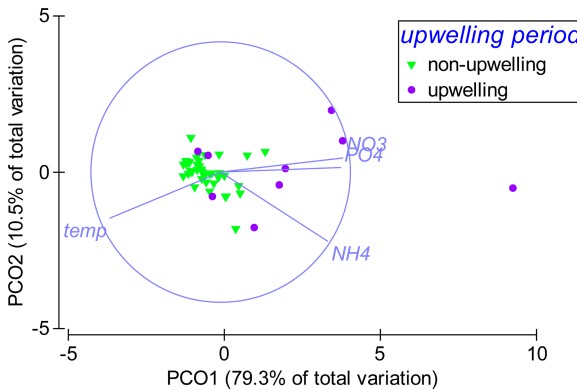

**Figure 6 Temporal pattern in environmental parameters.** Weekly sampling data from April 2013 to April 2014 were grouped by the factor upwelling period. The distance between data points reflects their similarity in environmental conditions (close = similar) and the separation along axes can be assigned to changes in environmental parameters (arrows). Sampling points in upwelling season are separated from the rest along PCO1, correlating highly with temperature ($r = -0.86$), phosphate ($PO_4$) ($r = 0.93$), ammonia ($NH_4$) ($r = 0.83$) and nitrate ($NO_3$) ($r = 0.95$).

which highly correlated with temperature ($r = -0.86$), $PO_4^{3-}$ ($r = 0.93$), $NH_4^+$ ($r = 0.83$) and $NO_3^-$ ($r = 0.95$) (Fig. 6). Environmental conditions were significantly different between the upwelling and non-upwelling period (PERMANOVA, Pseudo-$F(1, 45) = 19.87$, $P(\text{perm}) = 0.001$, perms = 999) and among individual months (PERMANOVA, Pseudo-$F(12, 34) = 2.7984$, $P(\text{perm}) = 0.006$, perms = 997). The temporal pattern in environmental parameters did not correlate to the pattern in community composition (BEST, Rho = 0.029, $P(\text{perm}) = 0.332$, perms = 999) and therefore did not explain any variance in the benthic community data.

## DISCUSSION

This is the first study to describe the variability in a benthic coral reef community in response to both an upwelling period and fluctuations in herbivore abundance in a high temporal resolution. The observations demonstrate the dynamic nature of the benthic community at the study site and provide detailed baselines for benthic community composition as well as fish and sea urchin abundances in coral reefs of the eastern tropical Pacific.

### Benthic community shift is mainly driven by high coral growth rates

An increase in absolute coral cover is important to retain a coral-dominated state in the face of environmental and anthropogenic disturbances, and therefore has been used by many studies to demonstrate reef recovery (*Hughes et al., 2010*; *Graham, Nash & Kool, 2011*). However, reef recovery cannot be measured based on coral abundance or growth rates alone, and instead requires measures of biodiversity, recruitment rates and colony size distribution—factors that were not assessed in the current study. Nevertheless, the rapid nature of the reef community dynamics observed at the present study site is exceptional. *Graham, Nash & Kool (2011)* reviewed recovery rates of absolute coral cover in all oceans and found mean annual rates of 3.56% (with a range from 0.13 to 12.49%). The annual

increase of almost 30% observed during the present study is among the fastest reported in the scientific literature, exceeded only by an annual live coral cover increase of >40% observed in the Great Barrier Reef following a coral bleaching event and subsequent bloom of the seaweed *Lobophora variegata* (*Diaz-Pulido et al., 2009*). The authors of this study attributed the unusually rapid regrowth of the branching coral *Acropora* (up to 100–200% within 6–12 months) to fast tissue regeneration of small remnants of unbleached live tissue, the high competitive capacity of the corals, and a seasonal dieback in the blooming seaweed. Similarly, the increase in coral cover during the present study did not involve new coral recruitment (no coral recruits were observed on settlement tiles in a parallel study; *Roth et al., 2015*), but rather was driven by the expansion of remaining live tissue over already existing dead skeleton and high linear growth rates (personal observation and parallel coral growth experiment, C Sánchez-Noguera, 2015, unpublished data).

Tissue regrowth over already existing coral skeleton seems to offer a rapid mechanism for coral cover increase, likely due to the fact that it does not require new skeletal growth and thereby reduces the energetic costs associated with calcification (*Diaz-Pulido et al., 2009*). In addition, the linear growth rates for *P. damicornis* and *P. elegans* in the Gulf of Papagayo are higher than reported anywhere else in the eastern tropical Pacific (up to $67 \pm 9 \text{ mm yr}^{-1}$ and $52 \pm 10 \text{ mm yr}^{-1}$, respectively; *Jiménez & Cortés, 2003*) and appear to be related to high nutrient and food availability during upwelling (*Glynn, 1977*; *Wellington & Glynn, 1983*; *D'Croz & O'Dea, 2007*).

## Herbivores likely control algal cover despite seasonal nutrient input

Previous studies in the Gulf of Papagayo reported increased abundances and biomass of *Caulerpa sertularioides* and *Sargassum liebmannii* in response to seasonal upwelling (*Fernández-García et al., 2012*; *Cortés, Samper-Villarreal & Bernecker, 2014*), suggesting that the reproductive cycle and benthic coverage of these fleshy macroalgae are primarily controlled by nutrient availability. In a study by *Fernández-García et al. (2012)* and in the current study, *C. sertularioides* cover on the reef decreased rapidly after the macroalgal's stolons and fronds turned white, which is a result of synchronous sexual gamete release. The spawning was likely triggered by changes in environmental conditions after the upwelling peak, when cover and density of the algae had been highest. Despite the pronounced decrease in *C. sertularioides* after the spawning event in April 2013, macroalgal cover at Matapalo only increased moderately during the following upwelling period in 2014, and therefore did not recover to the full extent of the previous year, despite the high availability of nutrients. The reproductive cycle of *C. sertularioides* may therefore be longer than one calendar year, or alternatively, the upwelling induced elevation in nutrient supply may not have been sustained long enough to induce a renewed macroalgal bloom.

Herbivores in coral reef ecosystems have long been recognized to serve as an important top–down control on the establishment and growth of algal communities (e.g., *Ogden & Lobel, 1978*; *Lewis, 1986*; *McCook, 1999*). In the Caribbean, grazing sea urchins such as *Diadema* can limit algal cover and thereby play a key role in increasing reef resilience by preventing phase shifts from coral to algal dominance (*Lessios, 1988*; *Knowlton, 1992*; *Hughes,*

*1994*). Models have shown that even a moderate sea urchin abundance of 1 ind. m$^{-2}$ may be sufficient to allow coral population recovery after disturbances (*Roff & Mumby, 2012*). The high abundances of *Diadema mexicanum* (around 5 ind. m$^{-2}$) at our study site suggest they may provide a key control of algal abundance also in the eastern tropical Pacific. We suggest that grazing by sea urchins was especially important in reducing turf algae abundance after a bloom at the beginning of the study period and was therefore responsible for the pronounced decrease in turf algae cover observed in June 2013. Abundances of *D. mexicanum* were positively correlated with turf algae cover, indicating migration or recruitment patterns of sea urchins according to food availability. By removing turf algae, *D. mexicanum* uncovered underlying crustose coralline algae and thereby likely facilitated the observed increase in live coral cover. High abundances of *Diadema* spp. can however also damage a reef by extensively eroding the carbonate substrate (e.g., *Ogden, 1977*; *Glynn, 1988*; *Eakin, 2001*). Abundances of sea urchins were above the threshold of 3 ind. m$^{-2}$ predicted to shift the reefs in the nearby Culebra Bay from a positive to a negative carbonate balance due to high erosion rates (*Alvarado, Cortés & Reyes-Bonilla, 2012*). The reef structure at Matapalo is homogenous and relatively flat, which could facilitate sea urchin erosion and thereby may reduce the structural stability of the carbonate framework.

Total reef fish biomass at Matapalo (51 ± 13 g m$^{-2}$) was comparable to baselines established in the south–western Atlantic (8–148 g m$^{-2}$; *Bruce et al., 2012*), but lower than in the Central Pacific (132–527 g m$^{-2}$; *Sandin et al., 2008*). Herbivorous fish biomass (16 ± 5 g m$^{-2}$) was lower than mean values for the Indo-Pacific, but higher than in the Caribbean (29 ± 4 and 9 ± 1 g m$^{-2}$ respectively; *Roff & Mumby, 2012*). Parrotfish biomass (11.4 ± 4.2 g m$^{-2}$) compromised most of the herbivorous fish biomass and was similar to mean values for the Indo-Pacific but much higher than those for the Caribbean (13.1 ± 2.4 and 6.7 ± 0.7 g m$^{-2}$, respectively; *Roff & Mumby, 2012*). The low biomass of top predators at Matapalo combined with the dominance of many small lower trophic level consumers could indicate degradation of the studied reef (*Sandin et al., 2008*). However, the quantification of fish in a relatively small and shallow reef area in the present study excluded large bodied predators and schools of medium sized fish which were present in the deeper and more exposed areas of the bay (personal observation of white tip reef sharks and schools of jacks and trevallies). Accordingly, we observed only one third of total fish species recorded on rocky and deeper study sites in nearby Culebra Bay (*Dominici-Arosemena et al., 2005*). In the study by *Dominici-Arosemena et al. (2005)*, planktivorous and invertebrate feeding species were more abundant than herbivores, resembling a trophic structure more typical of temperate rather than tropical regions possibly due to the influence of seasonal upwelling. In contrast, the relative abundance of herbivores was high at Matapalo reef (26–51%), indicating that fish communities in the Gulf of Papagayo differ largely with respect to depth and substrate type.

A parallel manipulative study at Matapalo reef showed that herbivorous fish controlled turf and fleshy macroalgal growth on long-term exposed terracotta settlement tiles, even during the nutrient-rich upwelling period (*Roth et al., 2015*). In contrast, the cover and biomass of turf algae on short-term exposed settlement tiles nearly doubled during

upwelling in February and March 2014 in response to increased nutrient concentrations. This effect was independent of grazing pressure, suggesting nutrient availability plays a role in regulating turf algae settlement (*Roth et al., 2015*). Due to their high surface to volume ratio, turf algae may take up nutrients from pulse events, such as upwelling, faster than other macroalgae (*Carpenter, 1990*), and thus episodic nutrient input may particularly support turf algal production and recruitment (*Russ & McCook, 1999*). Turf algae are able to rapidly overgrow corals, and increased nutrient concentrations can further exacerbate their competitive superiority (*Vermeij et al., 2010*; *Haas, El-Zibdah & Wild, 2010*). In contrast, the present study presents evidence that herbivory can control algae cover in the established reef community, even in times of high nutrient concentrations.

Turf algae harbor high abundances of microbes, including potential coral pathogens, and may therefore increase the prevalence of coral diseases and further negatively impact coral health (*Sweet, Bythell & Nugues, 2013*; *Casey et al., 2014*). The prevalence of coral disease at Matapalo was not elevated in months with high turf algal cover (I Stuhldreier, 2015, unpublished data). It would however be interesting to assess the microbial diversity at the studied reef, as microbial communities can give insight into the health and degradation status of reef communities (*Dinsdale et al., 2008*; *Bruce et al., 2012*).

## Local reefs remain sensitive despite their adaptation to upwelling conditions

The continuous increase in live coral cover over the studied year suggests that the investigated reef patch was recovering from a disturbance event in the past. However, the increase in coral cover was not apparent for all parts of the 1.2 km long reef. Furthermore, rapidly increasing coral cover does not always indicate a healthy and resilient reef (*Wooldridge, 2014*). Repeated coral mortality events over the last decades caused by sedimentation (*Jiménez, 2001b*), harmful algal blooms (*Guzmán et al., 1990*) and El Niño warming events (*Glynn, 1990*; *Guzmán & Cortés, 2001*; *Jiménez et al., 2001*) appear to have prevented the coral community at Matapalo from increasing in diversity or developing a more resistant structure. High abundances of sea urchins together with poor cementation of eastern tropical Pacific corals may therefore reduce the stability of local reef frameworks (*Cortés, 1997*; *Manzello et al., 2008*; *Alvarado, Cortés & Reyes-Bonilla, 2012*). Since maintaining reef framework integrity is equally important as coral growth and recruitment in the recovery process after disturbances (*Endean, 1976*; *Baker, Glynn & Riegl, 2008*), local reefs in the eastern tropical Pacific will remain sensitive to natural and anthropogenic disturbances, despite high coral growth rates.

In conclusion, hard corals and CCA benefited from prevailing environmental conditions during the year of observation, while cover of turf algae and fleshy macroalgae declined. Herbivores likely controlled turf algae growth, thereby facilitating fast coral expansion. In contrast, cover of the macroalga *C. sertularioides* appeared to be controlled by inorganic nutrient availability. Contrary to initial expectations, the benthic community composition did not follow a seasonal cycle corresponding to the non-upwelling and upwelling period, but instead shifted from turf algae to coral dominance over the year of observation. Interannual variability in conditions and community responses are likely high and future

studies should address the hypothesis that local reef communities are well adapted to upwelling conditions. The observation that sea urchins largely controlled turf algae cover in the reef should be verified via exclusion experiments.

## ACKNOWLEDGEMENTS

We thank I Gottwald for assistance in the field and the Centro de Investigación en Ciencias del Mar y Limnología (CIMAR), Universidad de Costa Rica, as well as the RIU Guanacaste for logistical support. We also thank the academic editor Fabiano Thompson and an anonymous reviewer for their constructive comments on the manuscript.

### Funding

This work was funded by the Leibniz Association. The funders had no role in study design, data collection and analysis, decision to publish, or preparation of the manuscript.

### Grant Disclosures

The following grant information was disclosed by the authors:
Leibniz Association.

### Competing Interests

Ines Stuhldreier, Celeste Sánchez-Noguera, Florian Roth, Tim Rixen and Christian Wild are employees of Leibniz Center for Tropical Marine Ecology. The authors declare there are no competing interests.

### Author Contributions

- Ines Stuhldreier conceived and designed the experiments, performed the experiments, analyzed the data, wrote the paper, prepared figures and/or tables.
- Celeste Sánchez-Noguera and Florian Roth performed the experiments, reviewed drafts of the paper.
- Carlos Jiménez reviewed drafts of the paper.
- Tim Rixen, Jorge Cortés and Christian Wild conceived and designed the experiments, reviewed drafts of the paper.

### Field Study Permissions

The following information was supplied relating to field study approvals (i.e., approving body and any reference numbers):

Field work was conducted under permits issued by the National System of Conservation Areas (SINAC) of Costa Rica (permit No: 019-2013-SINAC).

### Data Availability

The raw data for this research is provided in Supplemental Information 1.

## Supplemental Information

Supplemental information for this article can be found online at http://dx.doi.org/10.7717/peerj.1434#supplemental-information.

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
