# Peer review of "Dynamics in benthic community composition and influencing factors in an upwelling-exposed coral reef on the Pacific coast of Costa Rica"

_PeerJ, doi:10.7717/peerj.1434_

## Round 0.1 · original submission · Major Revisions

Dear Authors:

Your MS has been revised by two experts. One indicated major and the other reject. Both point out to the very same problem that is sampling design (spatial and temporal). I strongly recommend that you re-write the study in order to highlight the baselines for those reefs, and major features, instead of making bold conclusions that are not supported by the data, unless the authors can incorporate datasets for additional years, and quadrats as pointed out by referee#2. Considering the baselines (e.g. benthos, fish, water quality) for these reefs, how do they relate to baselines of other reefs in the Pacific and Atlantic ? (refer to e.g. to Bruce et al. 2012, Dinsdale et al. 2008).

If the authors decide to insist on the current format of the paper, I am afraid the MS will be rejected in the next round of revision.

Sincerely.

Fabiano Thompson

Reviewer 1 ·

Basic reporting

The manuscript “Benthic community shift in an upwelling-exposed coral reef on the Pacific coast of Costa Rica” by Stuhldreier at presents in a fine scale the coral cover increase and turf algae cover decrease in a upwelling-exposed reef in Costa Rica. Authors infer that the herbivorous are the major control to turf algae cover. The presented data for one year is consistent and the text is well written. However there are some issues that should be addressed before the manuscript publication. With this being said, I recommend major revisions.

Experimental design

I am concerned about the seasonality perspective. Authors have sampled just in one year and infer that no season pattern was found. Besides the present data be consistent I suppose that would be required sampling during each season at least two times. Please consider to explain this.

I suggest to classify the seasons instead provide the sampling dates (e.g. Summer, Spring, Winter, Autumn?). This is what authors mean by seasons?

Validity of the findings

In abstract this statement: “Sea urchin abundances were high and controlled turf algal biomass” (lines 34-35) seems to be too conclusive. Authors have not performed manipulative experiments to conclude this. This is an inference and should be stated in this way. I also found sea urchins abundance decrease; just like turf algae (see my comments below). I would expect the opposite.

Authors provides total fish abundance and biomass (Fig. 4B and C), however there is no data regarding other trophic guilds besides “herbivores” presented in manuscript text, figures or supplementary material. The biomass inference was done using taxonomic information (lines 146-148): “Biomass was afterwards calculated from mean length of each size class and species- or family-specific Bayesian length-weight relationship parameters available on FishBase (Froese & Pauly, 2012)”. Maybe authors can include the results from other trophic guilds once it can be very informative about reef health (e.g. Sandin et al., 2008). If authors choose to not include these important results, please include a sentence in Methods section explaining how “total fish” biomass was calculated and why they choose to present results in this way.

Additional comments

The manuscript could be shorter. I suggest authors shorten the text.

I have few comments for the Authors:

Major:

I understand that the main finding of the article is that in this year reef community composition did not oscillate very much during the seasons. Instead authors found that coral cover in Matapalo reef in this year (2013/2014) has increased and turf algae cover has decreased, probably due to herbivory (mainly sea urchins). If my understanding is correct I would recommend authors to address this issue more directly, since they just have data from one year. Without temporal replicates (sampling in more than one season/upwelling event) is hard (impossible?) to infer the effects of season/upwelling events in reef community composition.

Authors suggested that herbivores are the major control to turf algae cover. However looking at Figure 4A I noticed that abundance of sea urchin Diadema mexicanun have decreased in June 2013. Looking at Figure 2A I noticed that turf algae cover started to drop in late June 2013. I would expect the opposite. Seems contradictory. What would be the cause of the decrease sea urchin abundance at this reef?

Maybe authors can analyze directly the possible influence of Diadema abundance on turf algae cover. It seems

Obviously “fish total” biomass is greater than parrotfishes biomass. This reviewer was curious to find out what was the fish species composition of this site. The diversity/evenness has changed during (the year/upwelling season)? Maybe authors would calculate some diversity indexes and include in results.

Relationship of microbes and turf and microbes and coral Disease.

Minor comments:
Abstract:
In the last paragraph of introduction section authors provided a very comprehensive list of objectives and key questions. I suggest including a list of statements in abstract answering these questions, for example: “We found (i) that coral cover benefit from... ; (ii) …; (iii)”.

Introduction:
Lines 44-46: Improve this sentence. I suggest explaining that reef benthic communities are controlled by biotic and abiotic factors; authors would use the same factors as examples but please add references. Anthropogenic disturbances may affect these factors (e.g. removal of herbivores, load of nutients).

Methods:

Lines 113-121: This description looks as results. I suggest to remove it.

Line 165: I suggest moving this first sentence to the end of this sub-section.

Results:

I suggest author change subtopic titles answering introduction questions (affirmative sentence), i.e. what was the main finding in each subtopic.

I suggest including the statistical test names (i.e. PERMANOVA...) when presenting the results (e.g. line 206).

Lines 214-215: This sentence illustrates the need to include the raw fish data in SI table: “which was due to high abundances of Chromis atrilobata (Fig. 4b)”.

Lines 217-218: What fish species/trophic guild have contributed to the total fish biomass?
“Total fish biomass on the reef was significantly elevated in November and December 2013 compared to the following three months (Fig. 4c; F(4,20) = 14.729, P < 0.001).”

Line 242: How authors have categorized “environmental conditions”. Please add a sentence in Methods section.

Discussion:

Discussion section is too long. Please focus on the main findings and discuss it more directly.

I also suggest changing discussion subtopics to more meaning affirmative sentences.

Lines 249-250: Authors have sampled in only one upwelling event. I would change “seasonal upwelling” in manuscript to “upwelling event”.

Lines 254-255: the sentence “the scleractinian coral Pocillopora spp. benefited most from prevalent conditions in the year of observation” means that these corals were the most abundant? Not clear.

Lines 254-258: Repetitive from results section. Authors could remove or shorten these sentences.

Lines 276-277: This sentence seems to contradictory to the beginning of discussion section where authors states:
“Findings hint to a highly dynamic benthic community, which was however not directly influenced by the observed seasonality in environmental conditions during the study period.”

Lines 283-285: Nutrients can also influence positively turf algae cover (e.g. Vermeij et al., 2010). Authors should focus on the possible mechanisms on how turf was controlled in this sampling site during this year. I suggest raising some hypothesis to be further tested.

Line 318-320: Sometimes I found confusing sentences reporting others results. It is not totally clear if there was this study or previous. Make sure to state this clearer.

Line 341: Please change the word “prefer”. “Occur” maybe is more appropriate. This paragraph is bit repetitive (results). Please consider to remove it.

Lines 371-373: Again, when authors cite Roth et al 2015 work is not clear with was authors results or not. Please make sure to state sentences like: “In previous studies…”

Lines 378-379: Please consider to rephrase this sentence. Something like: “Here we show evidences of herbivory controlling turf algae cover, even in high nutrient condition”.

Lines 383-388: Hard to get the message, please consider to improve these sentences. Also “marginal reefs” was first referred here. What would be it? If is so important, please consider to add a small sentence in introduction or in Study site subsection (Methods section) explaining what this means. Matapalo reefs are marginal? Is not clear in manuscript text.

Lines 389-417: This paragraph is too long. Please consider shorten it.

Lines 418-431: The conclusion can be improved. Please consider addressing the questions raised in introduction here again and answer them. Those questions give an excellent guidance to readers follow authors’ ideas. I also suggest to address possible hypothesis to be answered in future studies.


Figures:
Fig 1 – I cannot see the “Upwelling influenced Gulf of Papagayo at the northern Pacific coast of Costa Rica” in fig 1a. I see clearly the Study Location in fig1a, I suggest authors add the upwelling information (maybe arrows?) in this map or change the figure legend. I also suggest including in legend explaining what “mwd” means. In what site the photograph was taken?

Fig 2 – It is not clear why the y-axis scale is so different. Please provide better explanation in figure legend. In text authors refer to fig 2a and 2b, however in figure 2 and in figure 2 legend there is no mention to a or b sub figures.

Fig 3 – Would authors use season names (summer, spring, winter, autumn)? Maybe would be clearer to readers.

Fig 5 – Both nutrients and temperature plots show clearly that the upwelling season occurred from February to mid-March. It is not clear why authors includes April and May in this season.

Fig 6 – Would authors use season names (summer, spring, winter, autumn)? Maybe would be clearer to readers.


References:

Sandin SA, Smith JE, DeMartini EE, Dinsdale EA, Donner SD, Friedlander AM, Konotchick T, Malay M, Maragos JE, Obura D, Pantos O, Paulay G, Richie M, Rohwer F, Schroeder RE, Walsh S, Jackson JBC, Knowlton N, Sala E. 2008. Baselines and Degradation of Coral Reefs in the Northern Line Islands. PLoS ONE 3.

Vermeij MJA, van Moorselaar I, Engelhard S, Hörnlein C, Vonk SM, Visser PM. 2010. The effects of nutrient enrichment and herbivore abundance on the ability of turf algae to overgrow coral in the Caribbean. PloS one 5:e14312.

Reviewer 2 ·

Basic reporting

The manuscript presents little potential of international impact mainly due to restricted size of studied area, problems with sample design and restriction of samplings of just one year.
The main findings is that coral abundance increase along the period of one year. Ok, and after? The abundance will increase or decrease? One year is a very restrict time for this kind of monitoring. Need at least two years of sampling.

Experimental design

The main problems are related to number of fixed quadrats and the absence of replicates and restriction of sampling of one year. Just five quadrats means one sample (mean + standard deviation). It is necessary at least more 10 or 15 quadrats, in order to have 3 or 4 group of samples with a minimum of distance one group of each other, representing replicas in the specific habitat of interest, shallow reefs.

Validity of the findings

Unfortunately, in my point of view the findings of manuscript will be valid only if more replicas of quadrats are included and samplings are extend for one more year.

---

## Round 0.2 · Minor Revisions

Dear Authors:

The manuscript has improved significantly. Before it can be accepted please address a few points raised by ref#1, i.e. improvements to the English language. I will then move forward and accept it ASAP.

Sincerely.

Reviewer 1 ·

Basic reporting

The manuscript has been greatly improved however some minor revisions should be done, especially in English language. I will point some sentences but the Authors may consider submitting the manuscript to an English revision service or a colleague. For instance there are too many “that”.

Experimental design

No Comments

Validity of the findings

No Comments

Additional comments

I recommend changing Results and Discussion sections subtitles to full sentences. For instance, what “High seasonality in water parameters” (line 238) means alone? The sentences should make sense by itself. Try to add the subsection message to each subtopic title.

Change sentences like “This study is the first to describe (…)” to something like “This is the first study to describe (…)”. (lines 258-259)

In lines 278-279 “our reef site” sound informal. Please change this sentence.

Subtopic “Herbivores control algal cover despite high nutrient input” (line 289) is too strong. Authors can suggest this but the results are not conclusive. More experiments/sampling is required to state this.

---

## Round 0.3 · accepted · Accept

Dear Dr. Ines Stuhldreier